# Successful Negative Pressure Therapy of Enteroatmospheric Fistula after Right Colectomy for Complicated Crohn’s Disease —A Proposal for a Three-Drain Wound-Separation Technique

**DOI:** 10.3390/medicina58020199

**Published:** 2022-01-28

**Authors:** Georgi Popivanov, Roberto Cirocchi, Dimitar Penchev, Kirien Kjossev, Marina Konaktchieva, Ventsislav Mutafchiyski

**Affiliations:** 1Department of Surgery, Military Medical Academy, “Sv. G. Sofiiski” Str. 3, 1606 Sofia, Bulgaria; d.k.penchev@gmail.com (D.P.); kirienkt@gmail.com (K.K.); ventzimm@gmail.com (V.M.); 2Department of Surgical Science, University of Perugia, 06123 Perugia, Italy; roberto.cirocchi@unipg.it; 3Department of Gastroenterology, Hepatology and Transplantology, Military Medical Academy, “Sv. G. Sofiiski” Str. 3, 1606 Sofia, Bulgaria; marina.konaktchieva@yahoo.com

**Keywords:** enteroatmospheric fistula, frozen abdomen, modified technique, staged wound closure

## Abstract

Enteroatmospheric fistulas (EAFs) are still the worst complication of the open abdomen. They lead to a significantly prolonged intensive care unit and hospital stay and to high mortality. Despite the various techniques described in the literature EAFs remain “a nightmare” for the patient, the surgeon, and the hospital. Here we describe a case of right colectomy for obstructing Crohn’s disease in a 26-year-old. On the 19th postoperative day, he developed a superficial EAF. Due to the frozen abdomen, neither resection of the anastomosis, nor implementation of the known techniques for treatment of EAFs were possible. This prompted us to modify the Pepe technique. The EAF was isolated from the upper and lower parts of the wound through deep-skin and subcutaneous sutures and the application of two small pieces of non-adherent plastic foil. The lower holes of a single drain, put through a piece of black foam, were placed over the fistula. The upper holes, which were enveloped with the foam, remained in contact with the wound. The drain was connected to a negative pressure of 125 mmHg. NPWT (negative pressure wound therapy) was also applied by two separate sponges and drains in the upper and lower part. The mainstay of EAF treatment is the isolation of the EAF from the abdominal cavity and subcutaneous tissue, supported by control of the sepsis and adequate nutrition. The proposed technique is applicable in cases with a single, superficial EAF on the background of the frozen abdomen with minimal lateral fascial retraction. As of today, due to the rarity of the condition and lack of randomized trials, EAFs still represents a unique challenge often requiring improvisation.

## 1. Introduction


*“Surgery, like music, and Jazz music, in particular, is more than a series of tones set in time, and although the rules are followed, every patient presents different challenges.”*
J. Martellucci [1].

Enteroatmospheric fistulas (EAFs), are the worst complication of the open abdomen notorious as “the nemesis of the open abdomen” and a “logistic catastrophe”. They lead to a significantly prolonged ICU and hospital stay, high mortality, low rate of primary fascial closure, and increased hospital cost with an average USD 412,000 per case [2,3,4]. The absence of the fistula tract, in contrast to the enterocutaneous fistula, hampers spontaneous healing. Despite the numerous techniques described in the literature EAFs remain “a nightmare” for the patient, the surgeon, and the hospital [5,6]. 

Herein, we describe a modification of the technique of Pepe et al. with a wound separation in a high-volume EAF as a sequel to an anastomotic leak after right colectomy [7].

## 2. Case Report

A 26-year-old man was admitted due to obstructive Crohn’ disease located at the ileocecal area and a skip obstructive lesion of the right flexure led to a significant weight loss (50 kg in one year). He was malnourished, with a BMI of 18.1, serum albumin 26 g/L, and hemoglobin 10 g/dL. After five days preoperative nutrition and correction of the homeostasis, he underwent right hemicolectomy with side-to-side antiperistaltic ileotransverse mechanical anastomosis. On the 9th postoperative day, he was reoperated due to signs of peritonitis. Intraoperatively, a small abscess near the colonic stump with an intact anastomosis was found. Of note, there were severe adhesions, which precluded not only the resection of the anastomosis but also the fashioning an ileostomy. The abscess was drained, the patient recovered and was discharged. Ten days later, he was readmitted again due to wound sepsis and dehiscence (5–7 cm). Intraoperatively, there was severe wound infection with fascial dehiscence and a frozen abdomen without signs of fistula, so NPWT was implemented. On the next day, a bowel fistula was manifested at the site of anastomosis without visible orifice (Figure 1). Neither resection of the anastomosis, nor implementation of the known techniques for treatment of EAFs was possible.

The frozen abdomen prompted us to modify the Pepe technique for fistula isolation. The wound was separated into three parts. The EAF was isolated from the upper and lower parts of the wound through deep-skin and subcutaneous sutures and the application of two small pieces of non-adherent plastic foil. The lower holes of a single drain, put through a piece of black foam, were placed over the fistula, whereas the upper holes, which were enveloped with the foam, remained in contact to the wound (Figure 2). The drain was connected to a negative pressure of 125 mmHg. NPWT was also applied by two separate sponges and drains in the upper and lower part (Figure 3 and Figure 4). The fistula output was an average of 500 mL per day. After the abatement of the infection, on the sixth day, a staged closure of the abdomen by a skin suture only was performed. Negative pressure remained only at the fistula site (Figure 5 and Figure 6). On the 26th day, the negative pressure was removed and a colostomy bag was placed (Figure 7). Almost complete healing was observed after 67 days (Figure 8).

## 3. Discussion

EAFs are heterogeneous and occur in a different background (patient general condition and underlying disease led to laparotomy) so good knowledge is crucial for successful management. The excellent review of Di Saverio et al. provides a comprehensive classification of EAFs and the known risk factors [5]. EAFs are classified as superficial or deep, according to the localization and high (>500 mL/day), middle (200–500 mL/day), or low volume (<200 mL/day) according to the amount of the discharge and whether it is single or multiple [5,8]. 

The resection of the fistula or proximal diversion is the best solution but is often impossible due to a frozen abdomen and lateral fascial retraction. In these cases, when possible, the resection through the “lateral surgical approach” described by Marinis and Demetriades, is the best option [6,9]. The approach to the EAF is through the circumference of the open abdomen wound or via vertical incision made 8–10 cm laterally. In all other cases, prevention of the contamination of the abdominal cavity and subcutaneous tissue through isolation of the EAF remains the mainstay of the treatment. The definitive reconstruction is indicated at a later stage, usually after 8–12 months [10].

Despite the several techniques described in the literature (NPWT, fistula-VAC, ring-silo, nipple-VAC, tube VAC, Pepe technique, floating stoma, primary suture, fistula plug or patch, fistula suspension, fibrin glue, acellular matrix, and pedicle flaps) no definitive recommendations can be proposed because EAFs are rare, and randomized control trials are impossible [5,6]. The significant variety of the EAF also contributes to the lack of a “universal” technique. Di Saverio et al. provide an algorithm for EAF management depending on the type of the fistula [5].

The proposed technique represents a modification of the “Pepe technique”. In contrast to the latter, only the tip of the drain is placed over the fistula, whereas the upper part, enveloped with foam, promotes healing. The direct suction through the drain allows application of higher pressure (125 mmHg) in contrast to 50–75 mmHg in Pepe et al. [7]. Another difference is the wound separation, which allows an independent staged closure. In contrast to “tube-VAC”, the drain is set to suction to avoid the spillage under the foam.

Last, but not least, general care remains an important part of the treatment because sepsis is the main cause of death in these patients. So the source control, meticulous skin care and broad-spectrum antibiotics are important parts of the treatment. Because of the catabolic state, high energy regimens with 30–35 kcal/kg/daily and 1.5 g protein/kg/daily are required with early enteral nutrition [11,12,13].

Of note, the proposed technique is applicable only in cases with a single EAF and minimal lateral retraction of the fascia (up to 5 cm). Although in the presented case the outcome was successful we are aware that this is improvisation. As of today, EAFs still remain a unique challenge, often requiring improvisation and surgical intuition [14]. As J. Martellucci wrote: *“Faced with the apparent paradox between evidence-based medicine and tailored surgery, the best treatment of every single patient often still needs the valuable intervention of improvisation to be effective. …* But *“you can’t improvise on nothing; you’ve gotta improvise on something. Improvisation cannot exist without experience and knowledge.”* [1].

## 4. Conclusions

The mainstay of EAF treatment is the isolation of the EAF from the abdominal cavity and subcutaneous tissue, supported by control of the sepsis and adequate nutrition. The technique described herein can be useful in cases with a single superficial EAF on the background of a frozen abdomen with minimal lateral fascial retraction.

## Figures and Tables

**Figure 1 medicina-58-00199-f001:**
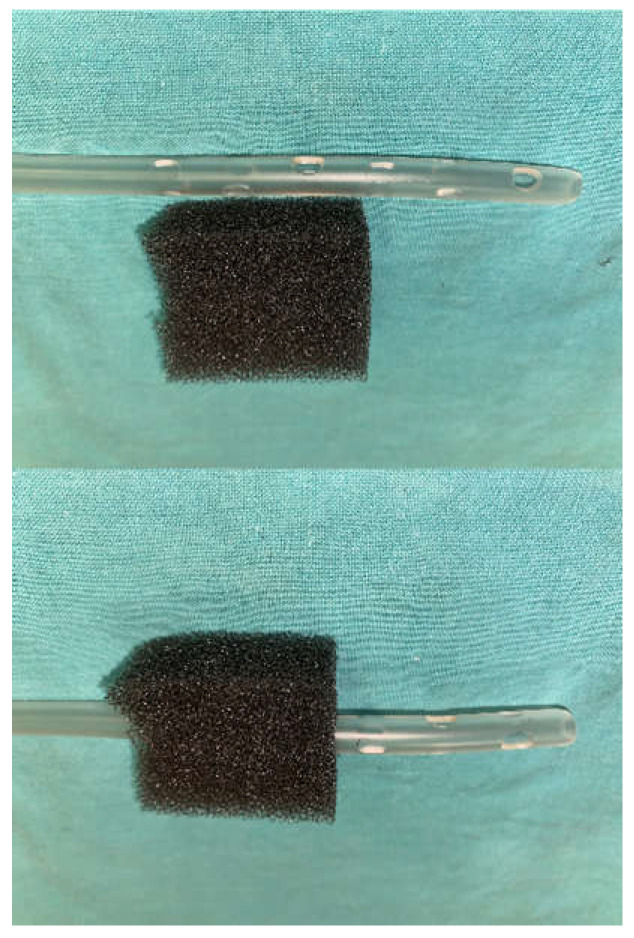
A view of the case after the first NPWT dressing.

**Figure 2 medicina-58-00199-f002:**
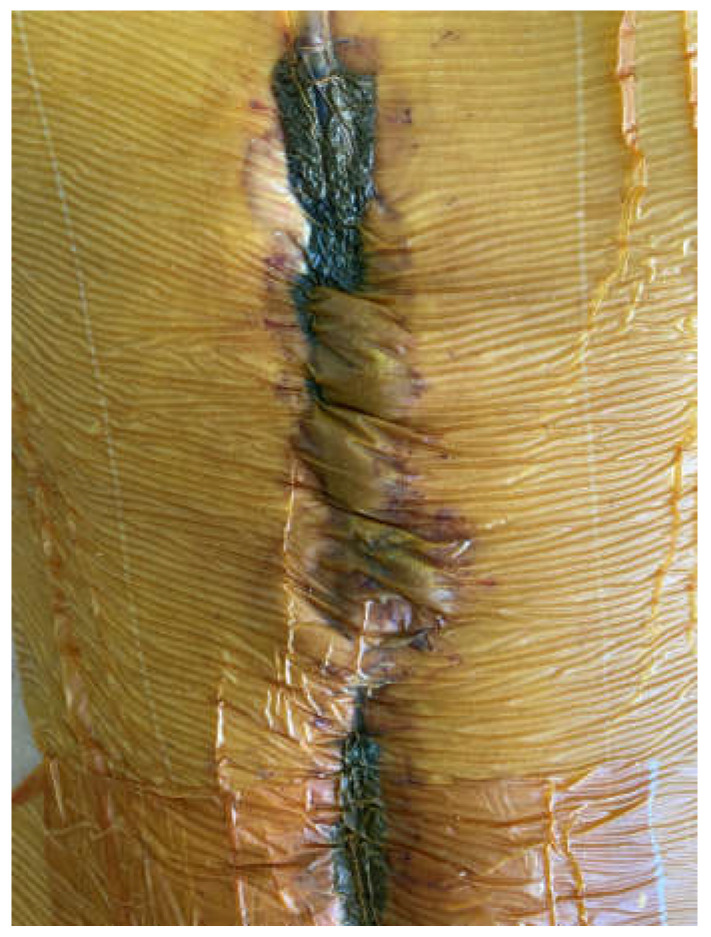
A view of the drain—the lower holes are put over the fistula, while the upper ones are enveloped with a black foam contacting the wound.

**Figure 3 medicina-58-00199-f003:**
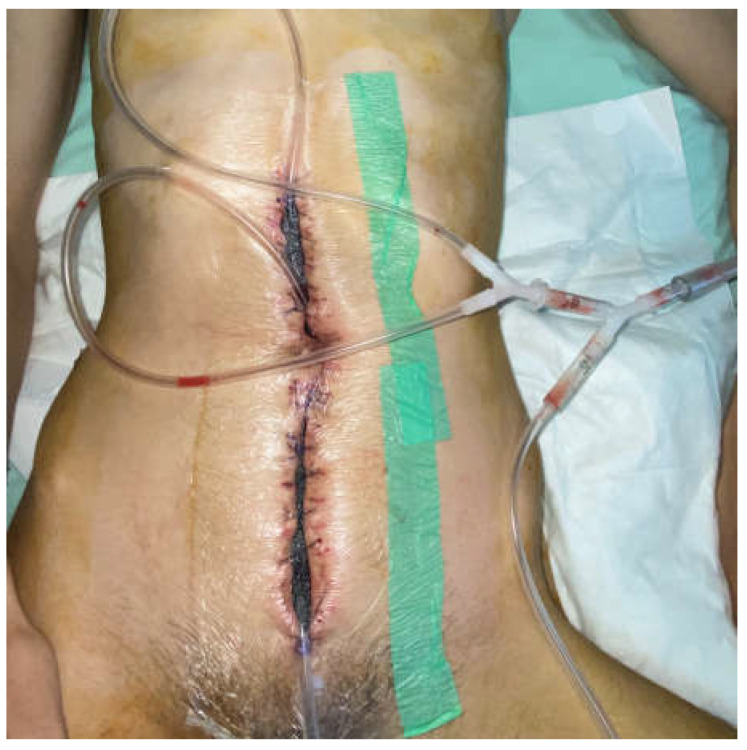
The technique with a suction drain in the fistula (in the middle) and separation of the EAF from the upper and lower part of the wound using two sponges and separate suction.

**Figure 4 medicina-58-00199-f004:**
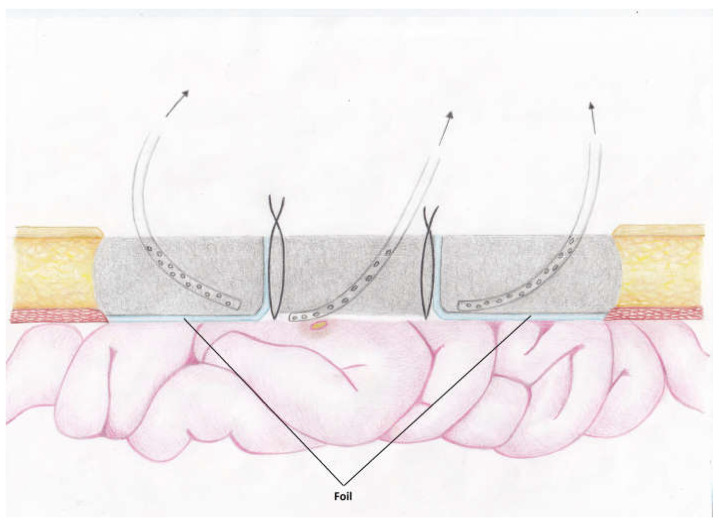
Schematic view of the proposed modification.

**Figure 5 medicina-58-00199-f005:**
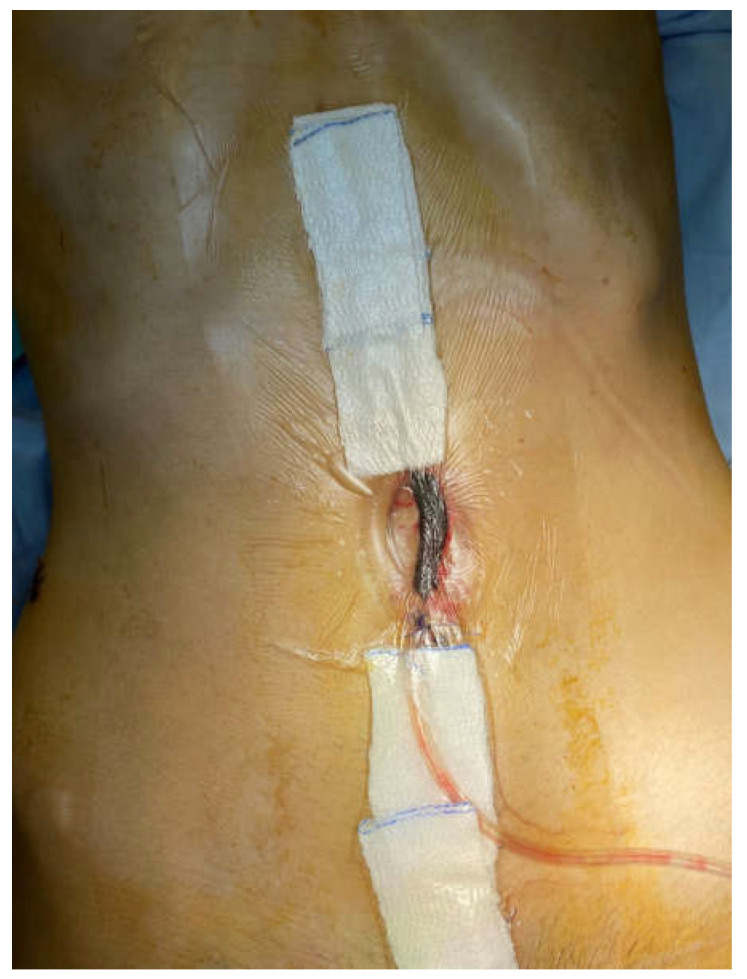
Upper and lower staged closure of the skin, NPWT at the site of the EAF (6 days after the start of treatment).

**Figure 6 medicina-58-00199-f006:**
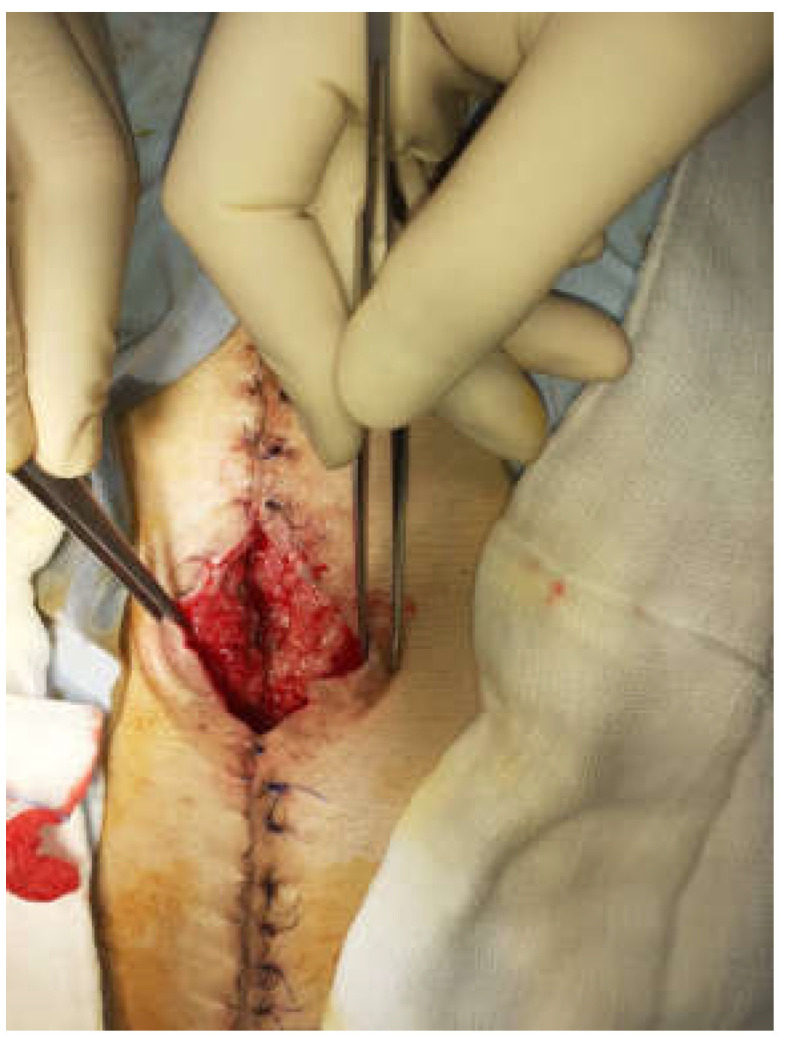
A view of the fistula site 12 days after the start of treatment.

**Figure 7 medicina-58-00199-f007:**
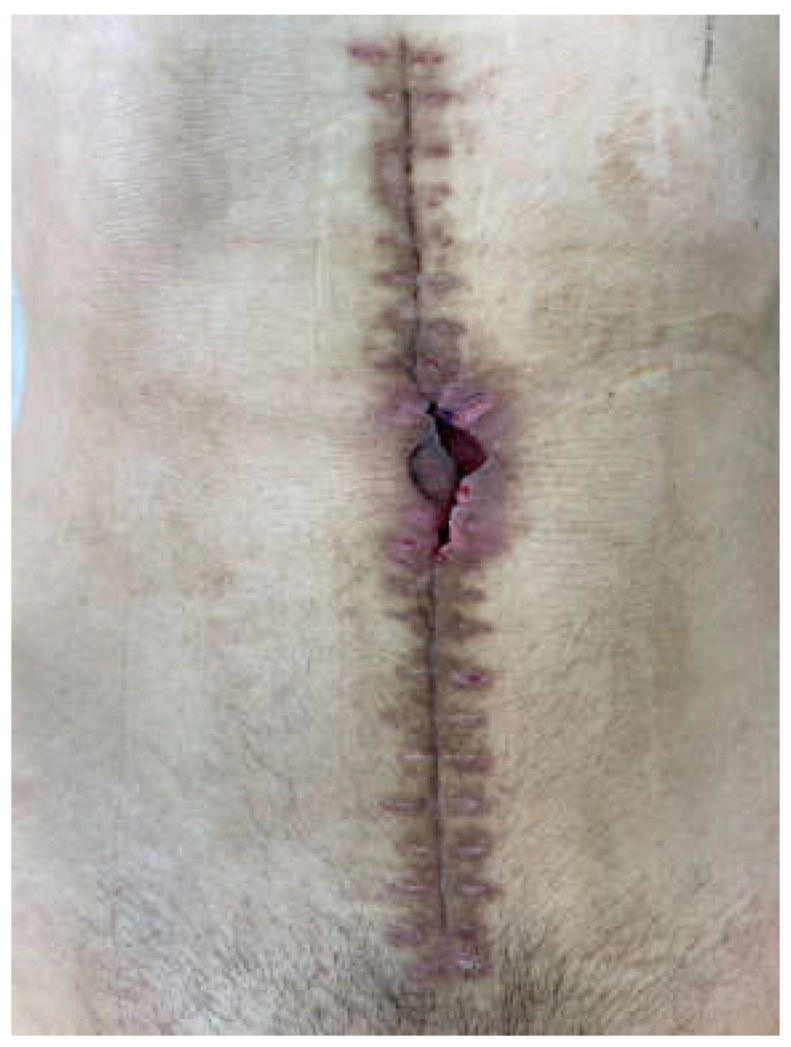
A view of the fistula site after 26 days, at that time colostomy bag was applied.

**Figure 8 medicina-58-00199-f008:**
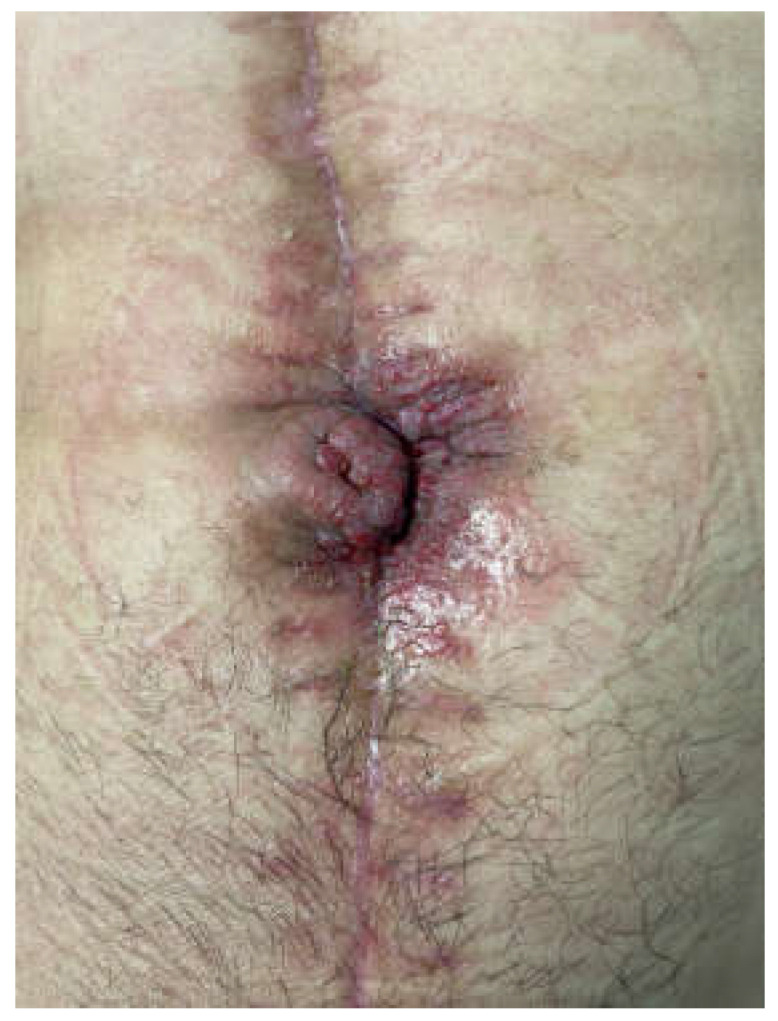
The fistula after 67 days after removal of the colostomy bag.

## Data Availability

Not applicable.

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
