# Peer review of "Successful Negative Pressure Therapy of Enteroatmospheric Fistula after Right Colectomy for Complicated Crohn’s Disease —A Proposal for a Three-Drain Wound-Separation Technique"

_medicina, 2022, doi:10.3390/medicina58020199_

Round 1

Reviewer 1 Report

Please review text for syntax and sentence structure.

This is a very interesting case report that is useful in describing a surgical technique that may be of benefit to surgeons who are confronted with these challenging cases.

Author Response

Done, Thanks for the remarks

Reviewer 2 Report

This is a really interesting paper of a disaster of the abdomen. Authors should clarify better their suggested technique: 1. is the drain with the sponge and the holes inserted in the middle of the wound and separate sponges on the upper and lower parts? Please provide a schematic depiction of this technique. 2. Are all connected in the same collector (as is shown on their photo? If yes, how did they measure output of the fistula? 3. No photo shows the colostomy bag applied on the middle of the wound were the fistula is. 4. Please show we’re the plastic coils are put.

Authors should explain why did they find a frozen abdomen after 10’days when peritonitis occurred. We all know that severe adhesions do not occur so early.

Reviewer 3 Report

As the problem of entero-atmospheric fistula is current and interesting, this case report is also interesting as a modification of the known techniques and of course the good result.

My comment is only this:

I think that the lateral intervention of the fistula area in cases of Open Abdomen, as it is proposed by Demetriadis et all (Journal of Trauma 2003) and Marinis et all (Surgical Infections 2009) is interesting to be presented properly in the discussion because:

  • In cases where is feasible to be performed, is favorable to other techniques like ‘‘nipple technique’’ or ‘‘floating stoma’’ concerning: a) hospital stay time, b) nursing facilities and c) money cost.

Author Response

Thank you for the remark

The lateral approach of Demetriades and Marinis was added as an option and references were reordered.